# Impact of the COVID-19 Pandemic Era on Residential Property Features: Pilot Studies in Poland

**DOI:** 10.3390/ijerph19095665

**Published:** 2022-05-06

**Authors:** Katarzyna Kocur-Bera

**Affiliations:** Faculty of Geoengineering, University of Warmia and Mazury in Olsztyn, 10-719 Olsztyn, Poland; katarzyna.kocur@uwm.edu.pl; Tel.: +48-89-523-4563

**Keywords:** COVID-19 pandemic, housing location, external and internal factors of property, changing preferences

## Abstract

Flats/houses in the COVID-19 pandemic era became the central place for living, working, learning, studying and entertainment. According to Maslow’s pyramid, all the basic needs had to be satisfied within a single space, which caused a change in the importance of certain locational and physical features of the flat/house. This study aimed to investigate how the COVID-19 pandemic changed the perception of the environmental features and the physical features of flats/houses. The research material was obtained from a questionnaire study disseminated through different online channels. The study was conducted in Poland, and citizens’ preferences are linked to the prevailing spatial and socio-economic determinants. A group of respondents were presented with 23 features describing the location and 17 features describing the physical features of flats/houses. They were also asked questions about the level of satisfaction with the current location and housing features. The results were analysed, and the statistical significance of the difference in the perception of the location features and the physical features of the flat/house was verified using a Chi-squared test. The results demonstrated a change in the importance of certain attributes concerning both external and internal factors. The physical features of the flat/house appeared to be more important (from the respondents’ perspective) than the features related to the location, as most changes occurred in that group. The respondents indicated that access to medical care facilities had gained importance (+8%), while good access to public transport had declined (−9%). For the physical features of flats/houses, respondents from other countries also indicated the importance of other attributes, i.e., the floor area (+12%), number of rooms (+14%), additional rooms (+14%), and access to broadband Internet and digital platforms (+28%). The study showed that over 30% of respondents would change their flats/houses if their financial means permitted.

## 1. Introduction

There has always been the dream of creating an ideal space for an ideal society throughout history. Many events in the history of mankind changed the perception of an ideal space. Wars, epidemics, developments in technology, and a change in the economic conditions of a particular country contributed to changes in the perception of human needs and housing, work, and leisure patterns. We are currently witnessing one of the major world-changing events in recent decades. The COVID-19 pandemic has been responsible for 5.5 million deaths worldwide [1]. According to official data, there were 105,753 COVID-19-related deaths in Poland (see Figure 1). However, mortality statistics indicate a very large number of excessive deaths in Poland [2] due to (among other reasons) a lack of testing of certain people (with respiratory failure or pneumonia being indicated as the cause of death or other disorders). According to the report *Health at a Glance* [3], as regards the OECD countries, Poland is ranked second (after Mexico) in terms of the number of excess deaths per million inhabitants [2]. The actual number of deaths due to COVID-19 in Poland may, therefore, be many times higher.

The introduction of many restrictions due to the COVID-19 pandemic, e.g., limiting the possibility of moving around in an open space and the implementation of work performance, learning/studying, and entertainment via the Internet, forced people to redefine their goals and expectations. This concerns the way of performing work, learning, studying, spending free time, and people’s housing conditions [5,6]. The place of residence became the central place of life. The simultaneous use of a flat/house by all household members (school-age children, working adults) resulted in more intensive use of the flat/house [7,8,9,10,11,12].

For every human to achieve well-being, a group of needs must be satisfied. Maslow identified different levels of needs and the order in which they are to be satisfied. According to Maslow, the most basic level must be satisfied before a person could strongly desire secondary or higher-level needs (see Figure 2—Maslow’s pyramid). In the COVID-19 era, the flat/house became the place where people sought to satisfy all of their basic physiological human needs. 

Previous research into physical and psychological human well-being focused on achieving well-being along with economic well-being and on improving human health and physical and psychological well-being [16] outside the housing environment [17,18,19]. The restriction of free movement due to the COVID-19 pandemic reinforced concerns that well-being could not be achieved. The lack of proper space for working, learning, exercising, and privacy in a flat/house may increase stress levels and, consequently, affect citizens’ mood and health [20,21,22,23].

Therefore, the perception of current residential patterns may have changed during the COVID-19 pandemic due to lifestyle, work, and entertainment changes. These changes could have caused changes in flat/house patterns and affected housing demand. 

The location of a residential unit (external factors) and its physical features (internal factors) became crucial in satisfying daily human needs in the pandemic era. Before the pandemic, market analysts claimed that the property value/price is affected by the “location, location, and once again location” [23,24]. It was defined, inter alia, by the following: (1) the presence of amenities and availability of services (e.g., shops, healthcare facilities, schools, pubs, parks, etc.); (2) quality of the neighbourhood (e.g., safety, aesthetics, green areas); (3) access to public transport or a road to enable moving around quickly. These components allow a broad range of human functional and recreational needs to be satisfied [9,23,25,26,27,28].

Besides the locational features, each residential unit has a unique set of physical attributes such as the floor area, age, number of sleeping rooms, outdoor private space, etc. [9]. The literature has many articles on the effect of these attributes on purchasers’ preferences [14,27,29,30,31,32].

Under the COVID-19 pandemic conditions, since daily life took place mainly in the flat/house, its physical features (internal factors) assumed particular importance. The number of rooms, floor area of the flat/house, and any additional space that can be adapted to purposes other than originally intended have gained more value. In the pandemic era, a winter garden, extension to a building, a shed, or even a balcony [9] became important in performing work at home (online), resting, and the need to feel isolated after many days of staying together with other household members. 

The COVID-19 pandemic also changed the perception of the place where work was performed and forced people to perform it under home conditions. Remote working merged the home and workplace [33,34,35]. This resulted in a growing demand for a space for online working at home, with features similar to a traditional office [36,37]. The attributes that became important include access to information and telecommunication technology [6,38], natural illumination, good thermal and acoustic insulation [6], and access to fresh air [39]. During the pandemic, access to communication technologies and digital platforms became an absolute necessity in all aspects of daily life. Meetings via Skype, Zoom, Microsoft Teams, etc., became one of the basic media which enabled working, learning, studying, resting, having fun, and satisfying social needs. 

People who spent most of their time in flats/houses and performed various activities paid particular attention to energy-efficient appliances (e.g., work/computer, heating/air conditioning, cooking, etc.), as they were used extensively during the pandemic. In the future, energy-efficient solutions may become more desirable, e.g., low energy consumption appliances, windows with double or triple glazing, equipment for electricity generation from renewable energy sources, etc. The COVID-19 pandemic also changed the perception of natural solar energy and ventilation [40] in houses and flats. As demonstrated by the study results, pleasant views from the windows and acceptable lighting levels (different in the living room and the sleeping room) became very important for stress relief [41,42]. Moreover, access to good quality air in residential areas and natural ventilation also promote health and improve well-being, both of which were worsened by COVID-19 [40,43].

Living under the new pandemic conditions became more difficult for people residing in city centres with limited access to green areas [44,45]. Such locations offer limited possibilities for satisfying recreational needs. People living in the suburbs (or in urban peripheries) usually have adequate greenery around their houses. However, this is often associated with limited access to other facilities and services (e.g., shops, takeaway meals, cinemas, and theatres). A question arises: did the COVID-19 pandemic change the social perception of essential spatial attributes (external factors) or the features of flats/houses (internal factors)? 

The research conducted to date into the effect of the COVID-19 pandemic on the property market has involved various research methods. Researchers have often based their inquiries on survey methods, in particular concerning the assessment of flat attribute social needs [46,47], the assessment of the property market in the pandemic era by business brokers [47], and the effect of the pandemic on global economy and properties [48]. There have also been studies concerning the effect of COVID-19 on changes in property prices and income [49]. The IMI index (a specific market intensity indicator provided by the Italian Revenue Agency) was estimated along with the economic model of Lotka-Volterra [50,51]. There is also a noticeable trend in the literature to compare the COVID-19 pandemic with extreme, abnormal events [52] and to adopt the research methods adopted in the field to analyse the impact of the COVID-19 pandemic on the property market. In particular, this concerns research into the changes in sale prices and rents by analysing historical economic data series [53], the use of the hedonic price method [54,55], which enables the assessment of the market value of a property while taking account of both the internal and external features of properties [56] to quantitatively determine the impact of extreme [57] and environmental [58] events.

The current study presents preliminary opinions on changes in Polish preferences regarding the perception of residential properties’ external and internal features induced by the COVID-19 pandemic. The analyses focused on the surrounding environment (external factors) in which a residential property is situated and its physical attributes (internal factors). Previous research mainly showed the need for multiple functions to co-exist in the flat/house [34,59,60] during the COVID-19 era. The current study extended this theme to include spatial/locational determinants of the property surroundings. Research into changes in social preferences regarding the features of a flat and its surroundings in the COVID-19 pandemic era is of great importance. The pandemic brought humanity to a standstill. The place of residence became very important, as the restrictions on mobility forced an enormous number of people to work, learn/study, and be entertained in one place, with the participation of all flat/house residents. This helped the public become aware of the surrounding space and the importance of environmentally friendly solutions in flats/houses, which alleviate human stress and promote well-being. As shown by [40,61], the promotion of passive strategies in the design of flats/houses contributes to sustainable environmental development and helps combat climate change and meet ambitious energy efficiency targets. The current study fills the knowledge gap in terms of human preferences concerning the features of residential properties and their surroundings, taking into account the effect of pandemic conditions on the Polish property market.

The article is structured as follows: following the introduction and literature analysis (Section 1), the data acquired for the study and the research methods are described (Section 2). The results obtained from the questionnaire survey are then presented (Section 3) and discussed (Section 4), and the study is then summarised.

## 2. Materials and Methods

The investigation of changes in society’s needs regarding the location factors and physical attributes of the flat/house in the context of the COVID-19 pandemic was based on questionnaire surveys and their analysis. The procedure is provided in Figure 3. 

The questionnaire was developed exclusively for this study, using an online questionnaire and Microsoft Forms. The online study into preferences (SPS) was conducted from 1 December 2021 to 30 January 2022. The questionnaire was available online in the Polish language and was disseminated via several channels. The survey questionnaire was distributed by sending a link to academic institutions (universities, polytechnics, high schools, etc.), offices, institutions, property agencies, etc. Attention was focused on selecting units in which the employees/students address the issues related to property economics, property management, socio-economic geography, etc. The survey form was also distributed using social media (Facebook), particularly in discussion groups associated with property rental, sale of residential properties, and the construction of houses (e.g., “She’s building a house!”, “Loft interiors”, “The Barn project—the construction of house”, “if I were building for the second time...”, and others).

The study area covered all of Poland.

The selection of variables (external and internal attributes of residential properties) adopted for the study was based on an analysis of relevant literature describing the attributes of importance to the public concerning local property markets [34,39,59,60,62,63,64], and legislation regarding the effect of spatial attributes on the market value of properties [63,64,65,66].

The first part of the questionnaire asked the study participants about socio-demographic data, such as sex, age group, employment status, education, marital status, region of residence, characteristics of the place of residence location, and the type of flat in which the respondent spent the most time before the COVID-19 pandemic and during the pandemic (see Table 1).

The second part of the questionnaire (external factors) provided a list of external attributes linked to the location of the place of residence (e.g., the distance to the workplace, leisure and entertainment facilities, shops, etc.) and internal features (internal factors) directly linked to the flat/house (e.g., the floor space, number of rooms, additional space, view from the window, etc.). The external factors of location included 23 attributes, while the internal factor of the flat/house included 17 factors (see Table 2).

The final part of the questionnaire included questions concerning the level of satisfaction with the current place of residence. Moreover, the preferences regarding the declared change in both the residential location and the flat/housing type were investigated. 

The $\chi^{2}$ test was conducted based on the formula [67]:(1)$$\chi^{2}=\overset{i=1}{\underset{r}{\sum }}\frac{{\left(f_{i}-np_{i}\right)}^{2}}{np_{i}}$$
where:

$\chi^{2}$—Chi-squared test;

$f_{i}$—number of observed values from a particular interval;

$np_{i}$—number of *n* units which should be included in a particular interval.

In order to analyse the collected data and monitor their statistical validity, a method for identifying atypical observations and the missing data analysis was applied [67]. 

## 3. Results

The questionnaire concerning changes in social preferences regarding the place of residence caused by the COVID-19 pandemic was mainly completed by women (57.5%) in the age group of 18–54 years (73%). In total, 79.5% of respondents were workers/employees with higher education (71.5%). The highest percentage of the respondents were people in a civil partnership (74.5%), while 23.5% of respondents were single. The questionnaire was completed most often by inhabitants of the following voivodeships: Mazowieckie (13%), Warmińsko-Mazurskie (17%), Wielkopolskie (16.5%), and Podkarpackie (14.5%). In total, 47.5% of the examined respondents inhabited a city centre, 29.5% inhabited a city periphery, 11.5% inhabited a suburban area, 3.5% inhabited a rural area (central part), while 8% lived in a rural periphery. The respondents mainly lived in 2- or 3-room flats (44.5%) or detached houses (26%) (see Table 1). The largest group comprised people in a relationship with children, who accounted for 40% of the respondents. A total of 47.5% of them lived in urban areas. This may have indirectly affected the survey results because houses/flats situated in the city centre usually have poor access to green areas (parks, squares, forests, etc.) and people living in a relationship and having children frequently experience a shortage of residential space, often due to the high price of houses/flats in central locations.

The study found that the respondents recognise that the pandemic had an effect on their preferences regarding the characteristics of properties and their location/surroundings. COVID-19 changed their perception, as the lockdown forced everybody to operate under restricted lockdown conditions. The questionnaire also asked the respondents about the attributes of the surrounding environment of a property (external factors) which were important before the COVID-19 pandemic and how they changed during the pandemic. Attention was also drawn to the residential issue (internal factors) and its physical features. 

Regarding the external factors, the respondents indicated that 15 out of 23 attributes concerning the residential location had become less important to them, while 8 attributes had gained importance (see Figure 4). 

In the studied population, the greatest differences were noted for the perception of attribute E5, i.e., easy access to public transport or a road. In total, 9% fewer respondents indicated that this attribute was less important to them. The interest also decreased in the following features: [E10]—sentimental attachment to the neighbourhood (according to 7% of the respondents); [E7]—close proximity to school/kindergarten; [E9]—type of development; and [E21]—traffic intensity in the neighbourhood (a difference of 3.5%). Certain attributes gained in importance in the COVID-19 pandemic era. These included: [E4]—lots of greenery and squares in the vicinity of the place of residence; [E13]—proximity to a primary healthcare centre (outpatient clinic/hospital); [E16]—access to outdoor leisure and sports facilities; and [E22]—dispersed residential development. 

There were noticeable differences in the perception of certain external attributes by women and men (see: Figure 5) and among age groups (see: Figure 6). For example, during the COVID-19 pandemic, access to public means of transport became less important for women [E5], while for men, it became more important than before the pandemic. Similarly, the attributes of access to medical care facilities [E13], proximity to school/kindergarten [E7], and noise levels in the area [E21] were also rated differently. 

As regards the age groups, certain attributes also had different dynamics. For example, the feature E4 (greenery in the vicinity of the place of residence) was important for age groups of 25–54 years, E16 (access to leisure and sports facilities) was not important for people over 45 years of age, while E22 (dispersed development) was not important for the age group of over 55 years.

The statistical significance of the investigated differences in the perception of the external factors before and after the COVID-19 pandemic (for the entire population) was examined using the χ^2^ test (chi-squared test) at a significance level α < 0.10 (see Table 3). The differences in the perception of feature [E5], i.e., a reduction in the importance of the access to public transport means, and [E13], i.e., an increase in the importance of the proximity to a primary healthcare centre, are statistically significant. 

Regarding the internal factors (for the entire population), ten attributes gained importance, while seven became less important. Interest increased in the following aspects (see Figure 5): access to broadband Internet [I9] (by 28%); the number of rooms [I2] (by 14.5%); the presence of additional rooms [I8] (by 14%); brightness of rooms [I10] (by 7%); access to digital platforms [I14] (by 9.5%); attractiveness of the view from the window [I11] (by 7%); good thermal/acoustic insulation of the building [I12/I13] (by 2.5% and 5%, respectively); and energy-efficient equipment [I15]. In the COVID-19 era, the following features became less important: the appearance of the building [I3], the technology of building construction [I4], the building location zone [I5] (within the boundaries of a district/housing estate), the technical condition of a flat/house [I6], the arrangement of rooms [I7], the running costs related to the flat/house [I16], and the individual/special characteristics of the flat [I17] (see Figure 7). 

Similar to the external factors, there were noticeable differences in the perception of certain attributes of flats/houses according to gender or the age group. The effect of COVID-19 on the arrangement of rooms in the flat/house [I7] and the running cost of the flat/house [I16] was rated differently by women and men (see: Figure 8). The women believed that the arrangement of rooms in the flat/house should be different, while the men claimed that it had no effect. However, the men began to recognise the importance of rent, which was less important before the pandemic [I16]. 

The respondents’ responses broken down by age groups indicated that there were noticeable differences in the perception of the flat’s floor area [I1], the building location zone [I5], technical condition of the building [I6], the arrangement of rooms in the house/flat [I7], the brightness of rooms [I10], good thermal/acoustic insulation [I12/13], and the running costs per house/flat [I16]. For example, for people over 55 years of age, the flat’s floor area [I1] in the pandemic era and the brightness of rooms [I10] were not very important, while the technical condition of the building [I6] for the groups of 35–54 became more important (see: Figure 9).

Statistically significant differences in the change of the perception of internal factors (for the entire population), examined using the χ^2^ test (Chi-squared test) at a significance level α < 0.10, were found for six attributes (described by the symbols I1, I2, I8, I9, I14, and I17, see Table 3). 

Analysis of the respondents’ opinions on the level of satisfaction with the location of the current place of residents (external factors) and with the physical features (internal factors) of the flat/house (see Figure 10) showed that 47.5% of the respondents were satisfied with residing in the current location (external factors), 24% had a rather indifferent attitude towards the location of the current place of residence, while 28.5% of the respondents were rather dissatisfied. In total, 45% of the respondents did not want to change their district/housing estate location, while 29% declared the need for such a change. 

The physical features of the flat/house (internal factors) fully satisfied 55.5% of the respondents, 17% had no opinion on this issue, and 27.5% claimed that the COVID-19 pandemic had changed their perception of the needs to be satisfied by the flat/house. In total, 32.5% of the respondents declared a willingness to change the flat/house if their financial conditions allowed it, while 57.5% of the respondents declared that the emergence of the pandemic had not made them want to make such a change. Lastly, 10% had no opinion on this issue. 

## 4. Discussion

The COVID-19 pandemic took the whole world by surprise. Due to the introduced restrictions (limitations or a total ban on moving around within cities or districts), people were forced to reorganise how they worked, studied, or were entertained. Satisfying these needs within one’s own flat/house became extremely difficult. Before the outbreak of the COVID-19 pandemic, people rarely worked, learned or were entertained in their flats/houses [17,18,19]. According to Messenger, Gschwind [68], before the outbreak of the pandemic, remote work (regular or occasional) was performed by 23–25% of employees in Slovenia, Austria, Germany, and France, and by 7–10% of employees in Bulgaria, Italy, and Romania). The COVID-19 pandemic completely reversed this trend. The residential space overnight became the central space in which household members were forced to satisfy all their basic needs [13,14,15]. This situation necessitated a change in expectations regarding the place of residence location and the physical features of the flat/house. The most important features (also comprising the perception of the features of the property surrounding environment) included access to primary healthcare facilities, dispersed development, and greenery and squares in the vicinity of buildings. In the respondents’ opinion, only the first feature gained in statistical importance. Because the COVID-19 pandemic threatened the health and lives of all demographic social groups, getting to a healthcare facility quickly became more important for people. Before the increase in the risk of SARS-CoV-2 infection, healthcare facilities in close proximity to the home were mainly important for pensioners and families with small children [69,70]. The pandemic changed the significance of these places, as it affected the entire society. 

Maintaining a healthy environment during the isolation, when all functions were concentrated in flats/houses, resulted in the need to provide access to more than just four walls and the roof [71]. For this reason, dispersed residential buildings and the greenery around them became valuable. No possibility of moving around in such spaces, and the ban on entering parks and forests [72] (during the first phase of the pandemic in Poland, such a ban was introduced) highlighted the importance of a substitute for a free, green space in the vicinity of the place of residence [73]. According to research, contact with nature raises the happiness hormone levels in humans, which becomes particularly important during a pandemic [74].

In the respondents’ opinion, the COVID-19 pandemic changed the significance and importance of the attribute linked to the ease of access to public transport (E5). Since the pandemic necessitated working, learning, and studying and being entertained online in the place of residence, people stopped using public transport on a daily basis. It also became somewhat dangerous, as many people crammed within the small space or a bus or tram increased the risk of infection. People started to use means of transport that allowed them to move around alone or in a small group. During the initial phase of the pandemic, bicycles, electric scooters, mopeds, etc. [75,76] declined in popularity, yet significantly increased in popularity during the subsequent phases, when the bans on moving around in open spaces were removed.

In the respondents’ opinion, the perception of certain physical features of flats/houses (internal factors) changed as well. Attributes that had previously been of little importance took on greater significance. The most important difference indicated by the respondents was good access to broadband Internet (I9). Such a connection is made via copper telecommunication cables or optical fibre cables [77]. Before the pandemic, this feature was of little importance, as the Internet was mainly used outside the home: at work and in school via a mobile connection. The transfer of these functions to the flat/house necessitated good access to the Internet, which could only be offered by broadband Internet. The respondents also indicated the greater importance, during the pandemic era, of the flat’s floor area (I1), the number of rooms (I2), the presence of additional rooms (I8), and access to digital platforms (I14).

The first three features are related to the need to perform many functions simultaneously in a flat/house and the need for isolation. When living with other household members around the clock, every human feels the need to perform certain activities in isolation. Meetings or discussions held by employees or schoolchildren require isolation as they generate noise. Many smaller spaces divided by walls limit the spread of noise, which is why the respondents indicated that the number of rooms was more important in the pandemic era than the floor area of the flat/house. 

Open spaces with separated zones for cooking, eating meals, resting, or watching TV were very popular in Poland prior to the pandemic. In this regard, can the pandemic contribute to changes in the way zones are separated in flats/houses?

The property market responds very slowly to social needs, mainly due to the length of the investment cycle. Due to the acquisition of land intended for development, the preparation of a building permit design and the acquisition of the relevant administrative, and legal documents related to building permits and funding sources, the duration of a construction project can last from 2 to 4 years, depending on its scale. The reflection of social needs in the aspect of the features of the property’s surrounding environment and the physical features of the flat/house are perhaps more noticeable in the longer term. Did the pandemic change construction patterns? Should today’s investor, a purchaser of a flat/house, or a tenant always consider the possibility of another threat (a COVID-level pandemic) being likely to happen in the short term?

It is well known that environmental pollution poses a hazard to human existence, and the rapidly advancing climate change is also not encouraging [78], while the lessons learned from the current COVID-19 pandemic leave people fearing that the situation could repeat. Therefore, should new housing investments take these concerns into account? The literature offers solutions that promote residential properties with useful features in the pandemic era. Their task is to maintain the autonomy of both the household and the individual [35,59,60,78,79]. 

These solutions involve merging the shared spaces with personal spaces in residential buildings and are a response to the isolation which affected all humans during the pandemic. The proposed solutions concern solutions in residential buildings that include additional rooms for shared use. These rooms could be made available via a room rental system and provide additional space for teleworking [8]. Such promotion of the adaptation of space to various spatial and temporal needs at the scale of a building would be beneficial in terms of mitigating the conflicts that arise between functions at the flat scale (irrespective of whether they are basic, e.g., work performed by adults, children’s learning, or expressing oneself, e.g., resting). Moreover, it was also proposed to design private outdoor spaces, visually combined, such as balconies, that could satisfy the need for safe interaction with the surrounding community (visual or aural, e.g., from a balcony to a balcony, from the street to a balcony), while providing the need for “social expression” [80]. Taking these needs into account would also be of great importance to the health and well-being of the elderly people who live alone and are under threat of spatial and social isolation [81]. The presented proposals draw on the construction patterns promoted in Poland during the People’s Republic (the communist era). The spaces shared by all tenants of a building (e.g., laundries, drying rooms, rest and refreshment rooms, etc.) and the entrances to flats from the so-called gallery were intended to enhance the integration of tenants [81]. Given the existing circumstances, such solutions could fulfil their role.

In conclusion, when designing new flats, it is important to take advantage of the assets offered by the environment. As a result, the flats can offer better living conditions that promote residents’ health and well-being. Designers should prioritise certain recommendations [40], e.g., (1) the view from the windows should provide spatial diversity as well as privacy [82,83]; (2) diversity in terms of visual an thermal comfort, and adaptive floor plans to facilitate work, education, exercising, cooking, socialising, and on-screen entertainment [84]; (3) the arrangement of rooms in the flat, which can be adjusted to different uses and support the changing role of the house; (4) control of light, temperature, air, and noise in the rooms, adjusted to the human circadian rhythm [85,86]; (5) designing spaces that support positive social interactions—shared terraces, outdoor yards, and public lounges that support physical distance [87]; (6) possibility for the use of natural ventilation and thermal comfort at home; (7) connection with nature—outdoor public spaces need to be sufficiently large to contain buffer zones which will allow people to feel comfortable in social situations and which enable physical distance to be kept; (8) appropriate sizes and arrangements of units, which support the physical distance of at least two metres between individuals. 

The study presented in this article is very important for creating future housing policies. The results can support the decision-making processes of city and commune authorities as well as investors planning to implement development projects. The adjustment to new construction determinants, taking into account the pandemic conditions, can also have side effects. The adjustment of the existing houses/premises to satisfy the pandemic circumstances requires interfering with the building structure. This is not always safe (determined by the building design and construction technology) or feasible. The installation of additional telecommunication infrastructure, windows, partition walls, etc., requires the owners to have financial resources and interfere with the building structure. Consequently, buildings with a chaotic arrangement of windows (different colours, sizes, etc.) and numerous small spaces within the house/flat can be constructed. 

## 5. Conclusions and Research Prospects

The study aimed to investigate the respondents’ opinions on how the COVID-19 pandemic changed the perception of locational features and physical features of the flat/house. The study was conducted using an online questionnaire. A chi-squared test was performed, indicating whether the noted differences in perception were statistically significant. The study confirmed that these preferences had changed. Similarly, as indicated by the results of studies into preferences conducted among other societies, what became more important mainly included the number of rooms and the access to other additional rooms, broadband Internet, and digital platforms. What distinguished the Polish citizens’ preferences was that access to public transport was no longer as important as before the pandemic. The flat/house location used to be the main determinant that affected the demand for properties, yet the COVID-19 pandemic changed this perception. Will the residential property market respond to the new needs? This remains to be seen in the coming years. The research prospects of the author include more detailed studies and analyses concerning the effect of the COVID-19 pandemic on the property market, with a particular focus on housing location and social needs. The proposed methodology can particularly be applied in market analyses used by investors and the authorities developing housing policy.

## Figures and Tables

**Figure 1 ijerph-19-05665-f001:**
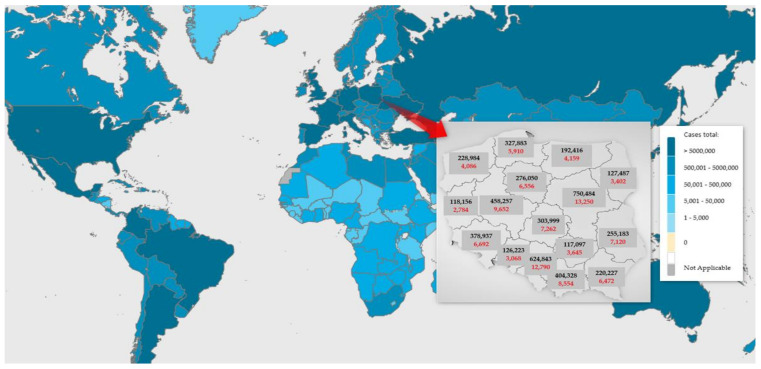
Map indicating the number of COVID-19 infections worldwide and in Poland (by voivodeships). Source: [1,4].

**Figure 2 ijerph-19-05665-f002:**
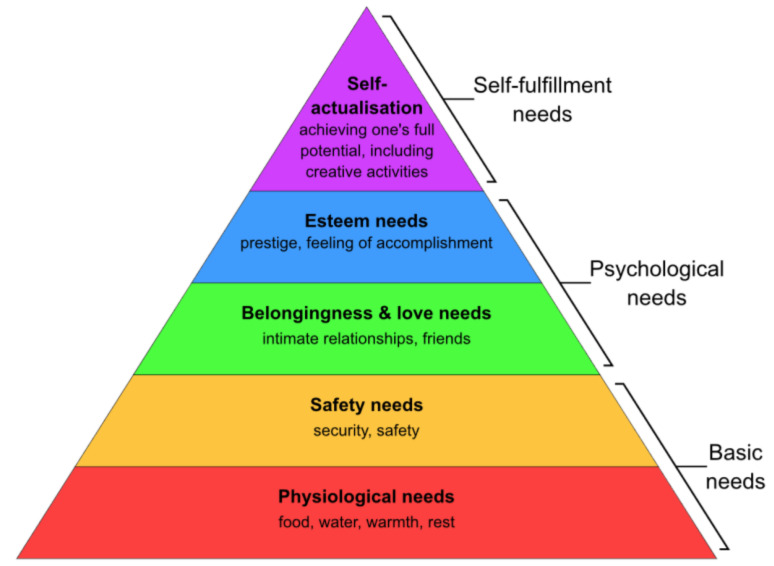
Maslow’s hierarchy of needs. Source: [13,14,15].

**Figure 3 ijerph-19-05665-f003:**
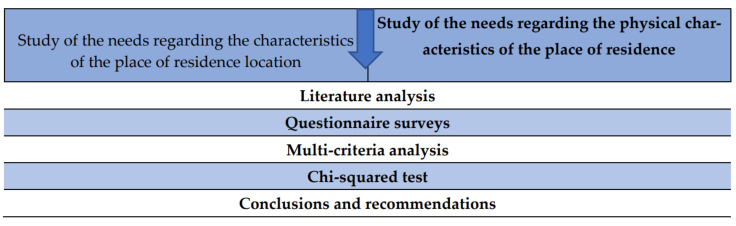
The study design.

**Figure 4 ijerph-19-05665-f004:**
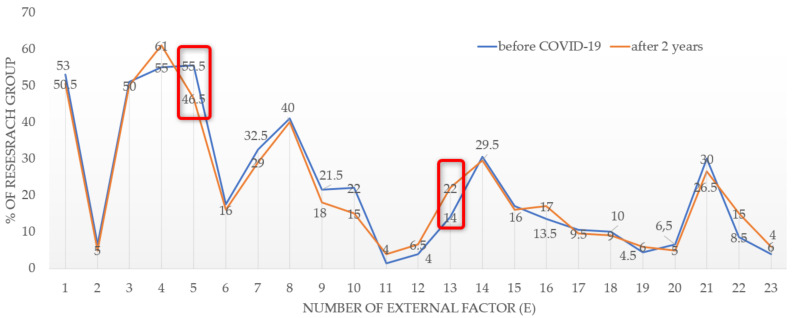
Changes in preferences in the residential location attributes after two years of COVID-19. Source: own study.

**Figure 5 ijerph-19-05665-f005:**
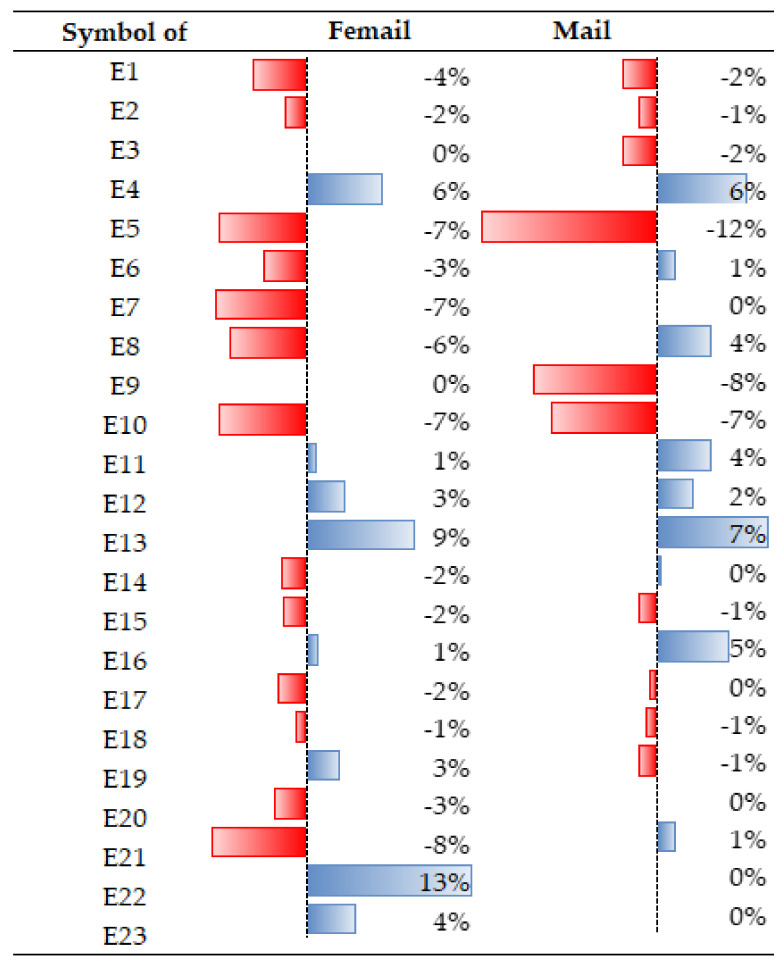
Differences in the perception of external factors, broken down by gender. Source: own study.

**Figure 6 ijerph-19-05665-f006:**
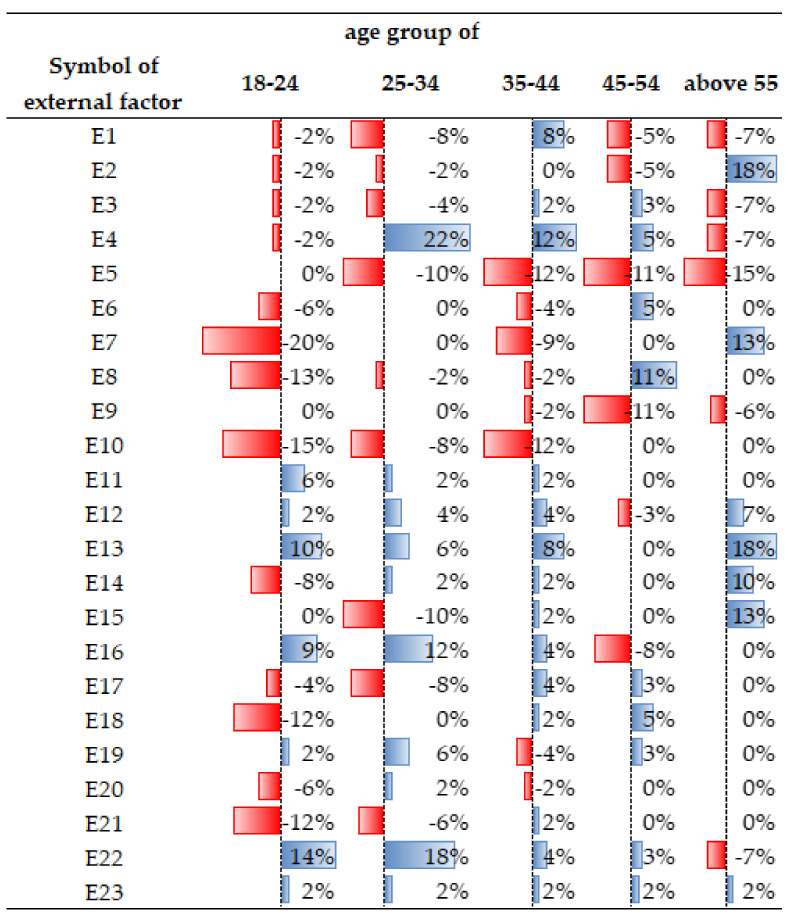
Differences in the perception of external factors, broken down by age groups. Source: own study.

**Figure 7 ijerph-19-05665-f007:**
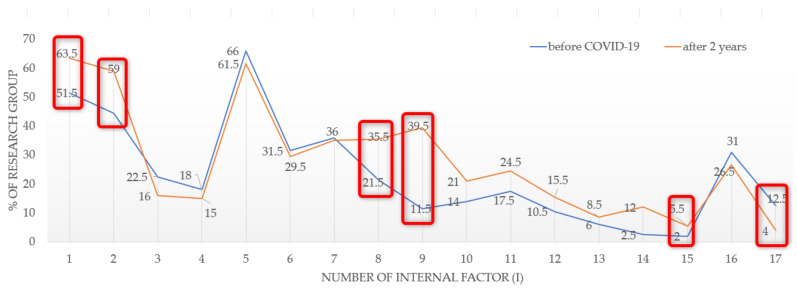
A change in preferences in the housing type attributes after two years of COVID-19. Source: own study.

**Figure 8 ijerph-19-05665-f008:**
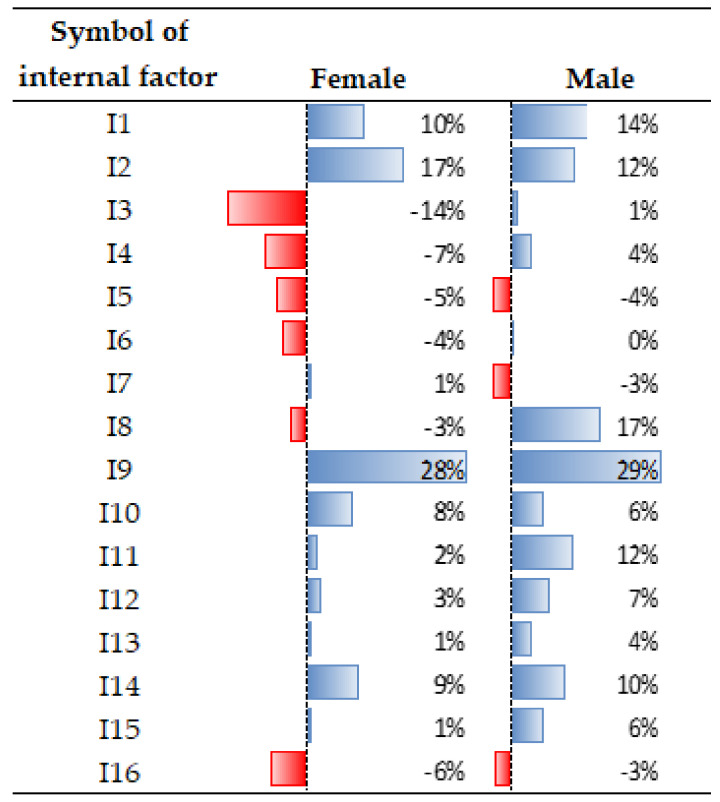
Differences in the perception of internal factors, broken down by gender. Source: own study.

**Figure 9 ijerph-19-05665-f009:**
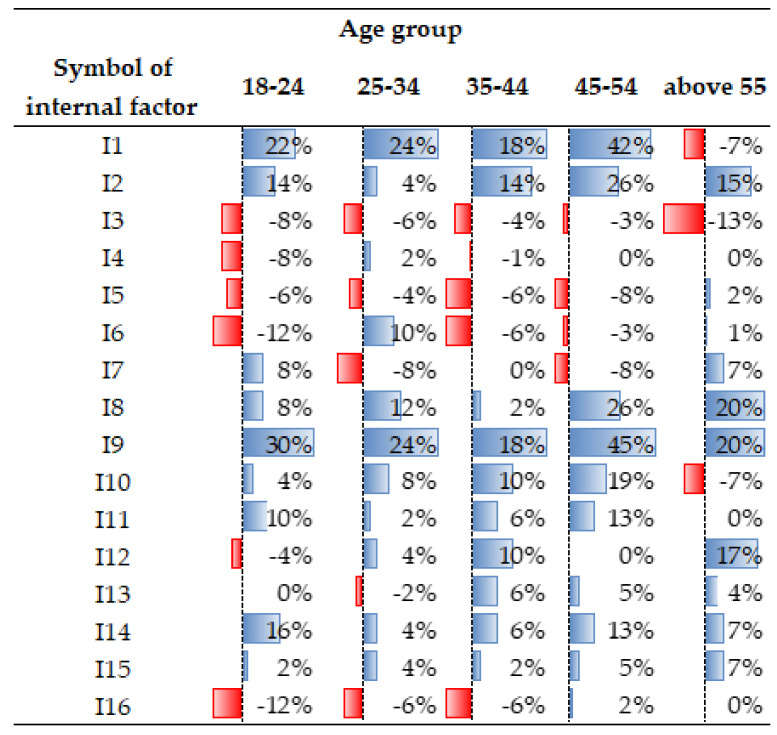
Differences in the perception of internal factors, broken down by age groups. Source: own study.

**Figure 10 ijerph-19-05665-f010:**
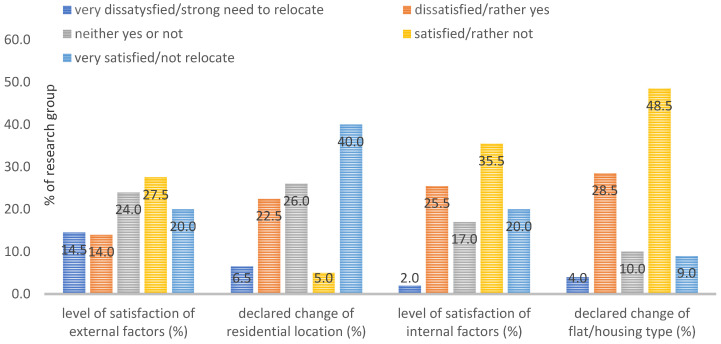
The level of satisfaction and the declaration of willingness to change the current residential location or housing type. Source: own study.

**Table 1 ijerph-19-05665-t001:** Test sample description.

Socio-Demographic Variables		*n*	(%)
Sex	man	84	42.0
	woman	115	57.5
	no response	1	0.5
Age group	18–24	50	25.0
	25–34	48	24.0
	35–44	49	24.5
	45–54	38	19.0
	55–64	11	5.5
	above 65	4	2.0
Employment status	employee/worker	159	79.5
	student/schoolchild	40	20.0
	non-working	1	0.5
Education	primary/vocational	1	0.5
	secondary	56	28.0
	higher	143	71.5
Marital status	single	47	23.5
	single/child/children	4	2.0
	in a relationship/no children	69	34.5
	in a relationship/with children	80	40.0
Place of residence region (voivodeship)	Kujawsko-Pomorskie	10	5.0
	Dolnośląskie	4	2.0
	Lubelskie	4	2.0
	Lubuskie	3	1.5
	Łódzkie	7	3.5
	Małopolskie	12	6.0
	Mazowieckie	26	13.0
	Opolskie	4	2.0
	Podlaskie	4	2.0
	Podkarpackie	29	14.5
	Pomorskie	10	5.0
	Śląskie	6	3.0
	Świętokrzyskie	4	2.0
	Warmińsko-Mazurskie	34	17.0
	Wielkopolskie	33	16.5
	Zachodniopomorskie	12	6.0
Characteristics of the place of residence location	city	95	47.5
	urban periphery	59	29.5
	suburban area	23	11.5
	village	7	3.5
	rural periphery (dispersed mode of settlement)	16	8.0
Housing type	a rented flat	17	8.5
	flat/1 room	15	7.5
	flat/2 rooms	43	21.5
	flat/3 rooms	46	23.0
	flat/4 rooms and more	13	6.5
	terraced house	7	3.5
	semi-detached house	9	4.5
	detached house	52	26.0
	other	4	2.0

Source: own study.

**Table 2 ijerph-19-05665-t002:** External factors of location and internal factors of the flat/house.

Symbol	Factor Description
E1	close proximity to the workplace
E2	trendy location (prestige of the place)
E3	easy access to small service facilities (greengrocer’s, grocer’s, chemist’s)
E4	lots of greenery, squares, lawns
E5	easy access to public transport (tram/bus/train)
E6	close proximity to entertainment facilities (restaurants, pubs, etc.)
E7	close proximity to school/kindergarten
E8	safety of the neighbourhood/district
E9	type of development
E10	sentimental attachment to the neighbourhood
E11	friends’/family members’ opinion on the location of the city/village/district
E12	information on the planned development of the neighbourhood (investments)
E13	proximity to a primary healthcare centre (outpatient clinic)
E14	access to supermarket/hypermarket
E15	access to parking spaces
E16	access to leisure and sports facilities (e.g., football pitches)
E17	appearance/aesthetics of the surrounding buildings
E18	proximity to playgrounds, outdoor gyms, skateparks, etc.
E19	accessibility of landscape architecture (benches, rubbish bins, fountains, monuments)
E20	the condition of pavements, curbs, driveways, roadways
E21	noise levels in the area
E22	dispersed development
E23	other
I1	floor area of the flat
I2	number of rooms
I3	appearance/aesthetics of the building/façade
I4	building construction technology (quality, lifespan)
I5	building location (zone, district, etc.)
I6	technical condition of the room/flat/house (utility systems, walls, floors, windows)
I7	arrangement of rooms in the flat/house
I8	presence of additional rooms (balcony, loggia, winter garden, shed, etc.)
I9	good access to broadband Internet
I10	brightness of rooms
I11	the attractiveness of the view from the window
I12	good thermal insulation of the building/flat/residential unit
I13	good acoustic insulation of the building/flat/residential unit
I14	access to digital platforms
I15	energy-efficient equipment
I16	running cost of the flat/apartment/house/residential unit
I17	other

Source: own study on [2,26,27,31,32,59,62,63,64,65,66].

**Table 3 ijerph-19-05665-t003:** Post-COVID-19 changes in external and internal factors (for the entire population).

Description	Factor Symbol	Differences (%)	t
close proximity to the workplace	E1	−2.5	1.58 *
trendy location (prestige of the place)	E2	−1.5	1.22 *
easy access to small service facilities (greengrocer’s, grocer’s, chemist’s)	E3	−1.0	1.0 *
lots of greenery, squares, lawns	E4	6.0	2.46 *
easy access to public transport (tram/bus/train)	E5	-9.0	3.0 **
close proximity to entertainment facilities (restaurants, pubs, etc.)	E6	−1.5	1.22 *
close proximity to school/kindergarten	E7	−3.5	1.87 *
safety of the neighbourhood/district	E8	−1.0	1.0 *
type of development	E9	−3.5	1.87 *
sentimental attachment to the neighbourhood	E10	−7.0	2.65 *
friends’/family members’ opinion on the location of the city/village/district	E11	2.5	1.58 *
information on the planned development of the neighbourhood (investments)	E12	2.5	1.58 *
proximity to a primary healthcare centre (outpatient clinic)	E13	8.0	2.83 **
access to supermarket/hypermarket	E14	−1.0	1.0 *
access to parking spaces	E15	−1.0	1.0 *
access to leisure and sports facilities (e.g., football pitches)	E16	3.5	1.87 *
appearance/aesthetics of the surrounding buildings	E17	−1.0	1.0 *
proximity to playgrounds, outdoor gyms, skateparks, etc.	E18	−1.0	1.0 *
accessibility of landscape architecture (benches, rubbish bins, fountains, monuments)	E19	1.5	1.22 *
the condition of pavements, curbs, driveways, roadways	E20	−1.5	1.22 *
noise levels in the area	E21	-3.5	1.87 *
dispersed development	E22	6.5	2.55 *
other	E23	2.0	1.41 *
flat’s floor area	I1	12.0	3.46 **
number of rooms	I2	14.5	3.80 **
appearance/aesthetics of the building/façade	I3	−6.5	2.56 *
building construction technology (quality, lifespan)	I4	−3.0	1.73 *
building location (zone, district, etc.)	I5	−4.5	2.12 *
technical condition of the room/flat/house (utility systems, walls, floors, windows)	I6	−2.0	1.41 *
room arrangement	I7	−1.0	1.0 *
presence of additional rooms (balcony, loggia, winter garden, shed, etc.)	I8	14.0	3.74 **
good access to broadband Internet	I9	28.0	5.29 *
brightness of rooms	I10	7.0	2.65 *
the attractiveness of the view from the window	I11	7.0	2.65 *
good thermal insulation of the building/flat/residential unit	I12	5.0	2.24 *
good acoustic insulation of the building/flat/residential unit	I13	2.5	1.58 *
access to digital platforms	I14	9.5	3.08 **
energy-efficient equipment	I15	3.5	1.87 *
running cost of the flat/apartment/house/residential unit	I16	−4.5	2.12 *
other	I17	−8.5	2.92 **
df			1
*p* < 0.1 **			2.7055
*p* < 0.05 *			3.8415

* confidence interval 0.90; ** confidence interval 0.95.

## Data Availability

Not applicable.

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
