# Peer review of "Impact of the COVID-19 Pandemic Era on Residential Property Features: Pilot Studies in Poland"

_ijerph, 2022, doi:10.3390/ijerph19095665_

Round 1

Reviewer 1 Report

The topic presented in the paper is current and interesting, the paper is well articulated and written, so minor revision are required before the publication.

In the Introduction Section to broaden the point of view on the effects of Covid-19 pandemic on the real estate market, it is recommended to widen the background to works related to assessment methodologies for the evaluation of the impacts of the COVID-19 Pandemic on the housing market demand  (there are some studies in the Italian reference).

In Section 2 Materials and Methods, it is advisable to deepen the description for the selection of the variables/factors and their attributes and to describe in a proper way the protocol adopted for the analysis of the data collected and for monitoring their statistical soundness. Also, it could be interesting to deepen the description of the dissemination of the questionnaire.

In Section 3 Results, it is advisable to add a brief reflection on how the characteristics of the respondents potentially influenced/deviated the results obtained through the questionnaire.

In Section 4 Discussion it is advisable to explain what decision-making processes the research carried out could support and what spillovers it can have in terms of interior space design.

It is advisable to modify the title of Section 5 in “Conclusion and research perspectives” and to add a small paragraph, highlighting the possible future applications of the proposed methodology.

Author Response

Dear Reviewer, 1

We thank you for the feedback provided and the suggestions, which I find relevant and useful to make this a clearer and stronger paper. All changes are marked in red. I have considered and acted upon each comment, and below are my responses and explanations of what has been done from my side.

Reviewer 1: Comments and Suggestions for Authors:

1: In the Introduction Section to broaden the point of view on the effects of Covid-19 pandemic on the real estate market, it is recommended to widen the background to works related to assessment methodologies for the evaluation of the impacts of the COVID-19 Pandemic on the housing market demand  (there are some studies in the Italian reference).

  • Thank you very much for the comments and suggestions of the Reviewer.  

The article takes into account the suggestions and in the "Introduction" section the description was extended to include the methodologies used in the real estate market - see l. 141-156. The article has the following description:

".... The research conducted to date into the effect of the COVID-19 pandemic on the property market has involved various research methods. Researchers have often based their inquiries on survey methods, in particular concerning the assessment of flat attribute social needs [46-47], the assessment of the property market in the pandemic era by business brokers [47] and the effect of the pandemic on global economy and properties [48]. There have also been studies concerning the effect of COVID-19 on changes in property prices and income [49]. The IMI index (a specific market intensity indicator provided by the Italian Revenue Agency) was estimated along with the economic model of Lotka-Volterra [50-51]. There is also a noticeable trend in the literature to compare the COVID-19 pandemic with extreme, abnormal events [52] and to adopt the research methods adopted in the field to analyse the impact of the COVID-19 pandemic on the property market. In particular, this concerns research into the changes in sale prices and rents by analysing historical economic data series [53], the use of the hedonic price method [54-55], which enables the assessment of the market value of a property while taking account of both the internal and external features of properties [56] to quantitatively determine the impact of extreme [57] and environmental [58] events….”

2: In Section 2 Materials and Methods, it is advisable to deepen the description for the selection of the variables/factors and their attributes and to describe in a proper way the protocol adopted for the analysis of the data collected and for monitoring their statistical soundness. Also, it could be interesting to deepen the description of the dissemination of the questionnaire.

  • Thank you very much for the comments and suggestions of the Reviewer. 

In the "Materials and methods" section, the description of the method of disseminating the questionnaire - years 196-204 :”…. The survey questionnaire was distributed by sending a link to academic institutions (universities, polytechnics, high schools, etc.), offices, institutions, property agencies, etc. Attention was focused on selecting units in which the employees/students address the issues related to property economics, property management, socio-economic geography, etc. The survey form was also distributed using social media (Facebook), particularly in discussion groups associated with property rental, sale of residential properties and the construction of houses (e.g. “She’s building a house!”, “Loft interiors”, “The Barn project – the construction of house”, “if I were building for the second time...”, and others)….”

and from where external and internal factors of residential property – l. 208-212 were obtained:

“…The selection of variables (external and internal attributes of residential properties) adopted for the study was based on an analysis of relevant literature describing the attributes of importance to the public concerning local property markets [34, 39, 59-60, 62-63, 64], and legislation regarding the effect of spatial attributes on the market value of properties [63-66]….”

  1. 3. In Section 3 Results, it is advisable to add a brief reflection on how the characteristics of the respondents potentially influenced/deviated the results obtained through the questionnaire.

- Thank you very much for the comments and suggestions of the Reviewer.

In the "Results" section, the descriptions of the research group and possible differences were extended - l. 251-257

“….The largest group comprised people living in a relationship and having children, who accounted for 40% of the respondents. A total of 47.5% of them lived in urban areas. This may have indirectly affected the survey results because houses/flats situated in the city centre usually have poor access to green areas (parks, squares, forests, etc.) and people living in a relationship and having children frequently experience a shortage of residential space, often due to the high price of houses/flats in central locations….”

Also, differences in the perception of external and internal factors were added, broken down by gender and age groups - l. 285-359.

  1. In Section 4 Discussion it is advisable to explain what decision-making processes the research carried out could support and what spillovers it can have in terms of interior space design.

- Thank you very much for the comments and suggestions of the Reviewer.

In section “Discussion” l.488-514 added: “….In conclusion, when designing new flats, it is important to take advantage of the assets offered by the environment. As a result, the flats can offer better living conditions that promote residents' health and well-being. Designers should prioritise certain recommendations [40], e.g. (1) the view from the windows should provide spatial diversity as well as privacy [82-83]; (2) diversity in terms of visual an thermal comfort, and adaptive floor plans to facilitate work, education, exercising, cooking, socialising and on-screen entertainment [84]; (3) the arrangement of rooms in the flat, which can be adjusted to different uses and support the changing role of the house; (4) control of light, temperature, air, and noise in the rooms, adjusted to the human circadian rhythm [85-86]; (5) designing spaces that support positive social interactions – shared terraces, outdoor yards and public lounges that support physical distance [87]; (6) possibility for the use of natural ventilation and thermal comfort at home; (7) connection with nature – outdoor public spaces need to be sufficiently large to contain buffer zones which will allow people to feel comfortable in social situations and which enable physical distance to be kept; (8) appropriate sizes and arrangements of units, which support the physical distance of at least two metres between individuals.

The study presented in this article is very important for creating future housing policies. The results can support the decision-making processes of city and commune authorities as well as investors planning to implement development projects. The adjustment to new construction determinants taking into account the pandemic conditions can also have side effects. The adjustment of the existing houses/premises to satisfy the pandemic circumstances requires interfering with the building structure. This is not always safe (determined by the building design and construction technology) or feasible. The installation of additional telecommunication infrastructure, windows, partition walls, etc., requires the owners to have financial resources and interfere with the building structure. Consequently, buildings with a chaotic arrangement of windows (different colours, sizes, etc.) and numerous small spaces within the house/flat can be constructed….”

  1. 5. It is advisable to modify the title of Section 5 in “Conclusion and research perspectives” and to add a small paragraph, highlighting the possible future applications of the proposed methodology.

- Thank you very much for the comments and suggestions of the Reviewer.

In section “Discussion” l.530-535 added: “….The research prospects of the author include more detailed studies and analyses concerning the effect of the COVID-19 pandemic on the property market, with a particular focus on housing location and social needs. The proposed methodology can particularly be applied in market analyses used by investors and the authorities developing housing policy...”.

Thank you in advance for your cooperation. I would appreciate it if you could send me further information.

  Author,

Katarzyna Kocur-Bera

Reviewer 2 Report

This study aims to investigate the changes in users’ perception of the environmental and physical features of their housing units. as a result of covid-19 pandemic. The study considered Poland as a case study. In general, the study is interesting. However, some changes are required to make it qualified for publication. These are as follows:

  1. The paper needs a comprehensive editing to improve English writing.
  2. The title is not clear. It needs to be re-written.
  3. The abstract should be more precise and to the point. It may be summarized.
  4. In the introduction, line 56: in-text referencing method should be consistent.
  5. Line 98: you need to say something about solar access and natural ventilation as essential housing design factors that improve well-being and reduce the potential of infection. See for example: Asfour, Omar S. (2022). Housing Experience in Gated Communities at the Time of Pandemics: Lessons Learned from COVID-19. International Journal of Environmental Research and Public Health, 19(4), 1925.
  6. Several studies could be found in the literature about the impact of the pandemic on housing design. For example:

Megahed, N.A.; Ghoneim, E.M. Antivirus-built environment Lessons learned from COVID-19 pandemic. Sustain. Cities Soc. 2020, 61, 102350

Bereitschaft, B.; Scheller, D. How might the COVID-19 pandemic affect 21st century urban design, planning, and development? Urban Sci. 2020, 4, 56

Peters, T.; Halleran, A. COVID-19 informed approach to examine apartment housing. Archnet-IJAR 2020, 15, 10–27

The introduction should highlight these studies and clearly discuss the research gap that would be bridged in the current study. 

  1. In Table 1: “a rented flat”: this could be one, two, three, or four rooms and more. Why did you ignored the area in the case of rented flats and emphasized on the area in the case of owned flats?
  2. Table 2 would look better if you removed the images and aligned the text.
  3. In Table 2: what do you mean by “room arrangement”?
  4. In Table 2: “I16” do you mean running cost?
  5. Line 168: Chi-squared test is a non-parametric test. Did you conduct data normality test to justify the use of this test?
  6. Figure 4: Add a title to the vertical axes. Also, make external factors numbering consistent with Table 2.
  7. In Table 3: the description column is empty.
  8. Is there any difference between the different groups in your sample? (e.g. male and female, age group, etc)?
  9. Support conclusion by data.

Author Response

Dear Reviewer, 2

We thank you for the feedback provided and the suggestions, which I find relevant and useful to make this a clearer and stronger paper. All changes are marked in red. We have considered and acted upon each comment, and below are our responses and explanations of what has been done from our side.

Reviewer 2: Comments and Suggestions for Authors:

R2: This study aims to investigate the changes in users’ perception of the environmental and physical features of their housing units. as a result of covid-19 pandemic. The study considered Poland as a case study. In general, the study is interesting. However, some changes are required to make it qualified for publication. These are as follows:

  1. The paper needs a comprehensive editing to improve English writing.
  • Thank you very much for the comments and suggestions of the Reviewer. The article has been re-edited. All corrections are in the text (green colour).
  1. The abstract should be more precise and to the point. It may be summarized.
  • Thank you very much for the comments and suggestions of the Reviewer. As suggested by the reviewer, the abstract was corrected and the most important survey results were added (in parentheses).

  1. In the introduction, line 56: in-text referencing method should be consistent.
  • Thank you very much for the comments and suggestions of the Reviewer. All comments have been taken into account.

Corrected as suggested by the Reviewer.

  1. Line 98: you need to say something about solar access and natural ventilation as essential housing design factors that improve well-being and reduce the potential of infection. See for example: Asfour, Omar S. (2022). Housing Experience in Gated Communities at the Time of Pandemics: Lessons Learned from COVID-19. International Journal of Environmental Research and Public Health, 19(4), 1925.

  • Thank you very much for the comments and suggestions of the Reviewer. All comments have been taken into account.

The description of the effect of the pandemic on internal factors was expanded and all suggested studies were included: l. 126-132 “….The COVID-19 pandemic also changed the perception of natural solar energy and ventilation [40] in houses and flats. As demonstrated by the study results, pleasant views from the windows and acceptable lighting levels (different in the living room and the sleeping room) became very important for stress relief [41-42]. Moreover, access to good quality air in residential areas and natural ventilation also promote health and improve well-being, both of which were worsened by COVID-19 [40, 43]….”

  1. Several studies could be found in the literature about the impact of the pandemic on housing design. For example:

Thank you very much for the comments and suggestions of the Reviewer. Other indicated test results were also taken into account.

  1. The introduction should highlight these studies and clearly discuss the research gap that would be bridged in the current study.

-              Thank you very much for the comments and suggestions of the Reviewer.

The element related to the importance of the topic was expanded : l.165-177 “…Research into changes in social preferences regarding the features of a flat and its surroundings in the COVID-19 pandemic era is of great importance. The pandemic brought humanity to a standstill. The place of residence became very important, as the restrictions on mobility forced an enormous number of people to work, learn/study and be entertained in one place, with the participation of all flat/house residents. This helped the public become aware of the surrounding space and the importance of environmentally friendly solutions in flats/houses, which alleviate human stress and promote well-being. As shown by [40, 61], the promotion of passive strategies in the design of flats/houses contributes to sustainable environmental development and helps combat climate change and meet ambitious energy efficiency targets. The current study fills the knowledge gap in terms of human preferences concerning the features of residential properties and their surroundings, taking into account the effect of pandemic conditions on the Polish property market….”

  1. In Table 1: “a rented flat”: this could be one, two, three, or four rooms and more. Why did you ignored the area in the case of rented flats and emphasized on the area in the case of owned flats?
  • Thank you very much for the comments and suggestions of the Reviewer.

Both the flat/house area and the number of rooms were taken into account in the research. The respondents indicated the importance of both internal factors.

  1. Table 2 would look better if you removed the images and aligned the text.
  • Corrected as suggested by the reviewer.

  1. In Table 2: what do you mean by “room arrangement”?

  • In Poland, flats are organized differently. For example, they have the so-called transition rooms (transition from room to room, and not from room to corridor), the kitchen can be a separate room or part of the living room. And there are also such solutions: a shared kitchen for two apartments (a post-communist relic). Arranging the room is therefore also an important factor.

10.In Table 2: “I16” do you mean running cost?

  • Yes, corrected after suggestion of Reviewer.

  1. Line 168: Chi-squared test is a non-parametric test. Did you conduct data normality test to justify the use of this test?

- The chi-quardrate test was presented to test whether the differences in the perception of the tested trait (external and internal) before the COVID-19 pandemic and after the 2-year duration of the pandemic are statistically significant. In a survey, this test is the most effective.

  1. Figure 4: Add a title to the vertical axes. Also, make external factors numbering consistent with Table 2.

  • Corrected after suggestion of Reviewer.

  1. In Table 3: the description column is empty.
  • Correceted after suggestion of Reviewer.

  1. Is there any difference between the different groups in your sample? (e.g. male and female, age group, etc)? Support conclusion by data.

  • Thank you very much for your valuable attention. The research was supplemented with these differences in the perception of external and internal factors by women and men and in age groups. The results were included in the article. See l. 285-352:

“…There were noticeable differences in the perception of certain external attributes by women and men (see: Figure 5) and among age groups (see: Figure 6). For example, during the COVID-19 pandemic, access to public means of transport became less important for women [E5], while for men, it became more important than before the pandemic. Similarly, the attributes of access to medical care facilities [E13], proximity to school/kindergarten [E7] and noise levels in the area [E21] were also rated differently........"

Thank you in advance for your cooperation. I would appreciate if you could send me further information.

  Author,

Katarzyna Kocur-Bera

Round 2

Reviewer 2 Report

Comments were fulfilled